



# Monitoring of Lower Thermospheric Neutral Density Variations Using Meteor Head Echoes

Devin Huyghebaert[1,2], Juha Vierinen[2,1], Björn Gustavsson[2], Ralph Latteck[1], Toralf Renkwitz[1], Marius Zecha[1], Claudia C. Stephan[1], J. Federico Conte[1], Daniel Kastinen[3], Johan Kero[3], and Jorge L. Chau[1]

[1]Leibniz Institute of Atmospheric Physics, University of Rostock, Germany
[2]Department of Physics and Technology, UiT The Arctic University of Norway, Norway
[3]Swedish Institute for Space Physics, Sweden

**Correspondence:** Devin Huyghebaert (huyghebaert@iap-kborn.de)

**Abstract.** Observations of neutral density in the mesosphere and lower thermosphere (MLT) region of the terrestrial atmosphere are important for understanding lower atmospheric, geomagnetic, and anthropogenic forcing. This study introduces a statistical method for measuring neutral density variations using an extensive dataset of meteor head echoes that were observed using the MAARSY high-power large-aperture (HPLA) mesosphere–stratosphere–troposphere (MST) radar. The method relies on
observing the mean geocentric velocity of meteor head echoes as a function of initial detection altitude and day-of-year. The meteor head echo catalog used contains 1.4 million meteor head echoes between 2016-2023. Neutral density variations are observed with a 3 day time and 2 km altitude resolution between 85-115 km. The measurements show variations in neutral density potentially due to geomagnetic and atmospheric events. Variations of 20-40% are common in the dataset, and agree with the magnitude of atmospheric neutral density fluctuations from an Upper-Atmosphere ICOsahedral Non-hydrostatic (UA-
ICON) atmosphere model run.

## 1   Introduction

Neutral density in the high latitude mesosphere and lower thermosphere (MLT) region is influenced by forcing from above and below and can act as a tracer for different geophysical phenomena. From above, the forcing is in the form of direct solar heating
as well as Joule heating due to auroral currents that close in the E-region ionosphere. The lower atmospheric forcings include phenomena related to gravity waves, planetary waves, and atmospheric tides. The MLT region is of particular significance for space weather, where variations in the neutral density can serve as indicators of, and will result in, variations in satellite drag and the atmospheric entry dynamics of spacecraft, space debris, and meteors. Small variations in the neutral density at the base of the thermosphere can correspond to large density perturbations at higher altitudes (e.g., Doornbos, 2012).



In this study we focus on the measurement of neutral density in the MLT. Previous researchers have used techniques involving specular meteor radars (Stober et al., 2012, 2014; Younger et al., 2015; Dawkins et al., 2024), high-power large-aperture (HPLA) meteor head echoes (Li and Close, 2016; Limonta et al., 2020), and incoherent scatter radars (Günzkofer et al., 2023; Thomas et al., 2024) for MLT neutral density measurements. Here we focus on providing a brief overview of the previous meteor measurement strategies for inferring details about the neutral density of the MLT region.

The monitoring of neutral density with meteor radars can be effectively separated into two categories based on the measurement technique. The two categories are as follows: radio scattering from the relatively stationary meteor trail and radio scattering from the fast-moving, localized meteor head plasma. The scattering from meteor trails require the signal to be approximately perpendicular to the meteor trail, providing a large radar cross-section and thereby allowing wide field-of-view measurements with relatively low power radar systems. Studies investigating height variations of the meteor trail and relating these variations to the neutral density can be found in Stober et al. (2012, 2014); Younger et al. (2015); Dawkins et al. (2024).

With this technique the height variations are determined as an average quantity with a neutral density isocontour assumed to follow this altitude variation. This provides a general overview of atmospheric neutral density variations, but provides minimal information about differences between altitudes for the same time.

Measurements of the meteor head plasma provide more details on the meteor ablation and trajectory, but require HPLA radars

(e.g., Chau and Woodman, 2004; Chau et al., 2007). The HPLA meteor head echo measurements can be used to determine the deceleration of the meteor for a given altitude (Li and Close, 2016; Limonta et al., 2020). This deceleration can be used with the scattering cross-section of the meteor and an accurate ablation model to estimate the neutral atmospheric density. There are some difficulties with this approach, including a list of assumptions and constraints on the fitting parameters provided in Limonta et al. (2020). Reported results compare relatively well with the MSIS neutral density model (Picone et al., 2002) and

can provide an altitude profile of the neutral atmospheric density.

We present a novel way of investigating the neutral density changes of the MLT region using HPLA meteor head echo measurements. This method is only possible due to the extensive dataset of meteor head echo measurements analyzed with the Middle Atmosphere Alomar Radar System (MAARSY) MST radar (Latteck et al., 2012). The analysis uses the three-dimensional trajectory and geocentric velocity of each meteor to determine relationships between the geocentric velocity,

initial radar detection altitude, and time of year of the meteors between multiple years. Details on the dataset and potential uses are provided in the following sections.

## 2    Method

MAARSY is a MST radar located on the island of Andøya, Norway (69.30°N, 16.04°E), and operates at 53.5 MHz (Latteck et al., 2012). The radar has a maximum peak power output of 433 × 2 kW and consists of 433 three-element crossed-Yagi

antennas with 16 receiver channels. These receiver channels can be used to determine the three-dimensional trajectory of meteoroids as they ablate in the terrestrial atmosphere through interferometry (Schult et al., 2013). In this study an analysis



similar to that of Schult et al. (2013) has been used for the detection and analysis of the meteor head echoes measured by MAARSY.

A meteor head plasma is different from the meteor trail plasma in that it is confined and very short-lived and corresponds
to the relatively high plasma density directly in the vicinity of the meteoroid as it ablates. By tracking the meteor head plasma with a radar it is possible to determine the velocity of the meteoroid. No perpendicularity condition to the trail is required for the head echo measurements, but a much larger radar aperture is usually required to observe the head echo due to the small target size of the dense plasma region surrounding the meteoroid.

The MAARSY radar has been previously used by Schult et al. (2017, 2018) for meteor head echo surveys. In the present
study we use meteor head echo measurements from 2016-2023 to investigate changes in the neutral density of the MLT region between years for the same day of year (DOY). A novel result is that we are able to obtain details on the neutral density variations as a function of altitude. Over 1.4 million good quality meteor head echo detections are included in this study. Note that different studies have investigated the height of meteor trails and meteor head echoes in the past, but to the authors knowledge this is the first dataset spanning 5+ years of consistent meteor head echo measurements. This allows comparisons
between the years to be made regarding the meteor ablation altitudes and meteor geocentric velocities.

The meteor head echo initial detection altitude is determined by several different factors: geocentric velocity, entry angle and position with respect to the radar beam, meteoroid composition and size, and atmospheric density (Vondrak et al., 2008). Two factors that we focus on are geocentric velocity $v_g$ and atmospheric density $\rho_a$, which affect the ablation-rate based on the equation (Vondrak et al., 2008),

$$70 \quad L\frac{dm}{dt} = \frac{1}{2}\pi R^2 v_g^3 \rho_a \Lambda - 4\pi R^2 \epsilon\sigma(T^4 - T_{env}^4) - \frac{4}{3}\pi R^3 \rho_m C\frac{dT}{dt} \qquad (1)$$

where $m$ is the mass of the meteor, $R$ is the radius of the particle, $v_g$ is the velocity of the particle, $\rho_a$ is the mass density of the atmosphere, $\Lambda$ is the "free molecular heat transfer coefficient", $\epsilon$ is the emissivity coefficient, $\sigma$ is the Stefan-Boltzmann constant, $T$ and $T_{env}$ are the temperatures of the surface of the meteoroid and the atmospheric temperature, $\rho_m$ is the mass density of the particle, $C$ is the specific heat, and $L$ is the latent heat of vaporization. Following Dimant and Oppenheim
(2017a, b); Sugar et al. (2018), we assume that the majority of the ionization produced in the ablation process is attributable to the meteoric material. The meteoroid geocentric velocity and the neutral atmospheric density are identified as the principal factors influencing the ablation rate and plasma density. A higher ablation rate will produce a larger plasma density, which then produces a larger radar cross-section. This suggests that meteoroids with higher velocity are, on average, detected at higher altitudes, and that an increase in atmospheric neutral density will result in the detection of meteor head echoes at a higher
altitude. This effect can be seen in Figure 1 as a correlation between initial detection height and $v_g$.

Due to the many factors in the ablation equations that are not known, we do not aim to estimate the absolute values of the neutral atmospheric density from the observed meteor ablation. Instead, our goal is to determine how the neutral atmospheric density varies between years at scales of a few days.

Throughout the year, the radiant distribution of meteors observed by a radar changes (e.g., Janches et al., 2006; Kero et al.,
2012). During the fall, the Earth's apex sources are at a higher elevation angle, allowing more high-velocity meteors to be





**Figure 1.** Histograms of the total meteor population as a function of initial detection altitude and geocentric velocity. An occurrence ratio contour of 0.1% of the total meteor count is included for the summer (red) and winter (blue) meteor distributions. The seasonal occurrence ratio plotted shows a difference in meteor counts in the summer (May 1 to September 1) and winter (November 1 to March 1).





observed, whereas during the spring months, much of the apex sources are below the horizon. The apex sources are produced from retrograde orbits of meteoroids, and the sources are at approximately 15 degrees north and 15 degrees south with respect to the Earth's ecliptic plane, in the direction of the Earth's orbital motion. During several time periods throughout the year, there are also meteor showers which are associated with larger meteoroids. These larger meteoroids can be detected at higher

altitudes. A few notable showers include the Eta Aquariids (≈ DOY 125 peak) and the Geminids (≈ 347 DOY peak). These factors influence the average detection altitude of a meteor. For this study we assume the meteor input function is relatively constant from year-to-year for the same DOY. This allows us to compare the average initial detection altitude from one year to another. The atmospheric density variations are then considered to be responsible for increases or decreases of the initial altitude for a given meteoroid velocity.

It is already shown in Figure 1 that there is a difference in the distribution of meteors for the summer and winter months. Meteors with larger geocentric velocities (> 40 km/s) tend to be detected at lower altitudes in summer than in winter. We do not attempt to separate differences in meteoric input from atmospheric variations as the cause of the change in distributions between seasons, as we do not need to when comparing the same DOY distributions between years. By subtracting the DOY data for each year from a background mean model we can observe variations in the geocentric velocities for the corresponding

initial meteor detection altitudes between years.

To confirm that trends in the meteor ablation characteristics follow trends in the neutral atmospheric density and to create the background model, the meteor head echo measurements were binned and averaged by DOY across the measurement period of 2016-2023. The results are presented in Figure 2, where the top panel shows the detection altitude of the meteors as a function of velocity and DOY, and the bottom panel shows the meteor velocity as a function of altitude and DOY. Also plotted in the

figure are the iso-altitude inverse density contours (top panel) and the iso-density altitude contours (bottom panel) from the Naval Research Laboratory Mass Spectrometer, Incoherent Scatter Radar Extended Model (MSIS) v2.1 atmospheric neutral density model (Picone et al., 2002; Emmert et al., 2021, 2022; Lucas, 2023). The meteor data was smoothed by applying a running 3-day average. We do not expect the meteor ablation characteristics to exactly match the MSIS model, only that there should be similar trends evident in both datasets.

Considering Equation 1, we may assume that the radar cross-section of a meteor plasma is $\propto \rho_A v_g^3$. We may therefore expect that there will be a decrease in the initial detection altitude of meteors at a constant velocity for a decrease in neutral atmospheric density. This can be clearly observed in the top panel of Figure 2 during the summer months, where the MSIS model contours are provided as the inverse of the density. Further, we can observe that the opposite is true for the smaller meteor velocities that reach lower altitudes before achieving an ablation rate detectable by the radar. These meteors are detected at

higher altitudes in the summer months than in the winter months.

We also provide a different way to illustrate the data in the bottom panel of Figure 2. Similar trends can be seen as those in the top panel, though bias from some of the meteor shower events are much more evident (e.g., Eta Aquariids in the Spring). These biases based on the seasonality of the data are a reason we only aim to investigate variations in the neutral density between years. By removing the average DOY trends from the initial data we can then investigate changes to the neutral density on





**Figure 2.** Meteor characteristics as a function of the DOY with a 3-day running mean. The mean initial meteor altitude with respect to the meteor geocentric velocity is shown in the top panel, with iso-altitude contours of the inverse density from the MSIS v2.1 model plotted in magenta. The bottom panel shows the mean geocentric velocity of the meteors with respect to the initial detection altitude, with iso-density contours from the MSIS model plotted in magenta.





time-scales of days. For ease of comparison with atmospheric models, we will use the bottom panel as the background velocity

as a function of altitude and investigate changes in the velocity of meteors for a given altitude between years.

The Upper-Atmosphere ICOsahedral Non-hydrostatic (UA-ICON) Model v2.1 (Kunze et al., 2024) is used in this study

to provide a relative magnitude verification of the results obtained. The UA-ICON model run is free-running from initial

conditions in 2018 starting from March 1. The horizontal resolution is 20 km and the output is taken from above MAARSY.

The solar radiation included in the model is from a monthly mean time series taken as an average over 3 solar cycles. No short

term geomagnetic effects are included. The vertical resolution is variable, though it is $\approx 1.5$ km in the altitude range of interest.

## 3  Results

The meteor geocentric velocity dataset was binned by initial altitude of detection and day with a 3-day moving average. The

background model for the DOY meteor velocities shown in the bottom panel of Figure 2 was then used to compare with.

To determine the change from the background model that can approximate the corresponding change in neutral density, the

following equation was used:

$$\frac{\Delta(v_g^3)}{v_{bg}^3} = \frac{v_g^3 - v_{bg}^3}{v_{bg}^3} \qquad (2)$$

where $v_{bg}$ corresponds to the geocentric velocity from the background model. The cubed geocentric velocity was used to relate

the change in velocity to the neutral density for a constant ablation rate based on Equation 1. Negative variations in velocity

therefore correspond to positive changes in neutral density. The results of this analysis are shown in Figure 3. Some data gaps

in the meteor head echo measurements are evident, though there are multiple years of data available for each of the days of the

year that go into the background model. Only DOY-altitude bins with more than 20 meteor head echoes detected are included

in the output, where the altitude bins are 2 km high.

There are some occurrences of cubed velocity variations from the background of 40+%. Though this appears as a very

large change in the neutral density, considering that the atmospheric density scales in an exponential manner and that the scale

height of the atmosphere corresponds to the temperature - which can vary by factors of 2 or more between seasons - the values

we observe here are quite reasonable. It can also be observed that the figure rarely shows neutral density variations at these

magnitudes, where it is more common to observe variation values between -20% to 20%. Further, to confirm that these neutral

atmospheric density changes between years are realistic, a UA-ICON model (Kunze et al., 2024) run has been examined and

shows similar magnitudes of changes in the MLT neutral density values. The output of this UA-ICON model with a similar

analysis to that performed on the MAARSY meteor head echo dataset is shown in Figure 4.

In Figure 4 clear oscillations can be observed when comparing the neutral atmospheric density for a given DOY from the

multi-year mean. The magnitudes are similar to Figure 3, though many of the oscillations are much more clear. This is not

unexpected. Comparisons between data and models will commonly produce different results. Even with the differences, there

are some trends that are evident between the datasets.





**Figure 3.** The variation between the cubed meteor geocentric velocity for a given initial altitude detection bin with a 3-day moving mean and the cubed average of the meteor geocentric velocity for the same centered DOY across all measurement years (bottom-panel Figure 2). The results are presented in this manner to approximately correspond to the neutral atmospheric density for a constant ablation rate (Equation 1). Red corresponds to an increase in the neutral density, while blue corresponds to a decrease. Only data with greater than 20 meteor head echoes detected in the altitude-DOY bin are shown. The black marks at 110-115 km correspond to SSW events and the green marks correspond to geomagnetic storm events with DST < -100 nT.




**Figure 4.** Presented is an UA-ICON Model run for MLT neutral atmospheric density comparisons with the meteor head echo data. A 3-day multi-year running mean around the DOY was taken for all years as the background model and the analyzed data was filtered with a 3-day running mean with the background model subtracted, the same as Figure 3.





## 4    Discussion

The data in Figure 1 and 2 are not new findings and are relatively well accepted in the community. Where we expand on the current state of the art is in Figure 3, where we remove the multi-year average for a given DOY from the data to only investigate variations between the years. For the MAARSY meteor data we examine the variation of $v_g^3$ based on the relationship between

meteor ablation rate, atmospheric density, and the geocentric velocity given in Equation 1. Changes in $v_g^3$ should approximately correspond to the atmospheric neutral density for a constant ablation rate and altitude. We consider the ablation rate at the initial detection altitude to correspond to a constant detection threshold with the radar.

Further examining the results, there are many different geophysical phenomena that can be linked to the results displayed in Figure 3. Here, we focus on some noticeable events and trends in the meteor head echo data and highlight some similarities

with the UA-ICON Model. Only a qualitative analysis is performed here to show that this method of determining atmospheric neutral density variations with meteor head echo measurements is viable for long-term monitoring of this region. As this is a difficult region to consistently measure for neutral density quantities, there are very few datasets available for comparison.

### 4.1    Sudden Stratospheric Warmings

Zhou et al. (2023) showed that there is a positive increase in the neutral atmospheric density at altitudes between 85 and 95 km

in the days after the onset of the SSW at a similar latitude to that of MAARSY in an epoch analysis. In this current study, we examine the neutral density variations at altitudes of 85 to 115 km, which is slightly above the altitude of the study by Zhou et al. (2023). From Sato et al. (2023), we can consider the following dates for major SSWs in the MAARSY dataset (YYYY/MM/DD format, DOY in parantheses): 2016/02/09 (40), 2017/02/01 (32), 2018/02/12 (43), 2019/01/01 (1), and 2021/01/05 (5). All of these dates show a relative decrease in the meteor geocentric velocity ratio in Figure 3 (highlighted by black marks at 110-

115 km), corresponding to a relative increase in neutral atmospheric density.

### 4.2    Solar Cycle Effects

The solar cycle will have an effect on the thermospheric neutral density, where during solar maximum it is expected that the density at a fixed altitude will be larger (Doornbos, 2012). This increase in the neutral density can be attributed to the increased temperatures, which cause lower, more dense layers of the atmosphere to expand vertically to higher altitudes. In addition,

enhanced dissociation and ionization of O2 and N2, for example, result in an increase of the neutral density (number of neutral particles per unit volume) In Figure 3, it can be observed that 2016 shows the largest neutral densities on average and it was during the decline of the previous solar maximum period. The solar minimum of 2019-2020 shows on average the lowest corresponding neutral densities, and we are beginning to observe increases in the neutral density again for 2023.



## 4.3 Geomagnetic Storms

Yamazaki et al. (2024) has previously investigated changes to the MLT region using SABER data and Hp30 geomagnetic activity index. Increases to the MLT region temperatures were evident during geomagnetic storm-time conditions. There was also a time of day dependence on the temperature variations.

Some dates where the DST was less than -100 nT are provided here, where this threshold is considered to be between moderate and intense geomagnetic storm-time conditions. A different threshold would either result in too many events for the purpose of this qualitative assessment, or not enough events for a proper determination of whether the data shows any trend. The dates selected to be examined are (DOY in parentheses): 2016/10/13 (287), 2017/05/28 (148), 2017/09/08 (251), 2023/02/27 (58), 2023/03/23 (82), 2023/04/23 (113), 2023/11/05 (309), and 2023/12/01 (335). These dates have in general a less clear signature in Figure 3 (highlighted by green marks at 110-115 km), though a further analysis is required to properly assess the impact of the geomagnetic storms at the location of MAARSY.

## 4.4 Potential Turbopause Monitoring

In the MAARSY meteor head echo data there is a consistent feature in the dataset where, at altitudes above 100 km in the summer and autumn, the neutral density variations are reduced between the years compared to other seasons and altitudes. If we compare with the UA-ICON model data in Figure 4, we can observe that there is also a reduction in the variability of the neutral density at these times and altitudes. It should be noted that this region is above the approximate altitude of the turbopause (e.g., Hall et al., 2016), which could be a reason for this reduction in variation.

## 4.5 Planetary Wave Activity

There are many instances of potential planetary waves in the MAARSY meteor head echo data. These are especially evident in the winter season with a periodicity of several days. Some examples of planetary waves in the data include the density variation oscillations during ≈ day 350 of year 2018 and 2020. In 2018 especially it can be observed that there is an altitude propagation component to the variations in the neutral density. It is clear that wave activity on multi-day scales is detectable with this method of analyzing the meteor head echoes for MLT investigations.

## 4.6 Closing Remarks

To close the discussion, we do not expect the UA-ICON model and MAARSY meteor head echo data to exactly match. The UA-ICON model is a free running model that produces a realization of atmospheric parameters derived from sea level forcing. For the density variations derived from MAARSY meteor head echoes in this analysis we have to assume that the neutral density is the primary cause of changing meteor ablation rates for a given altitude at the same time each year. It is sufficient to say that many characteristics between the model and MAARSY meteor head echo derived neutral density variations are similar. Further validation of the MAARSY meteor head echo data is planned, with the potential to constrain the neutral density values through the implementation of accurate meteor ablation models with the data.



## 5 Summary

Here we have shown a novel method to measure neutral density variations between years for the same DOY. This is possible due to the extensive dataset of over 1.4 million meteor head echoes during the years of 2016-2023 that have been recorded with the MAARSY radar system. The data provides MLT neutral density information with 2 km altitude resolution between 85-115 km and a time resolution of approximately 3 days. Many different geophysical phenomena are evident in the dataset, such as sudden stratospheric warmings, planetary waves, and geomagnetic storms. These will be explored further in the future. One of the valuable features of this analysis is the ability to provide an altitude profile to the neutral density variations which allows the altitude propagation of different perturbations to be detected.

We show that the fluctuation level of neutral MLT densities derived from MAARSY meteor head echo measurements are consistent with neutral density models. This novel method of monitoring the neutral density can therefore contribute to constraining the neutral density variations in models at the difficult to measure MLT region altitudes. Combining the neutral density variations from MAARSY with the SIMONe Norway (Huyghebaert et al., 2022; Jaen et al., 2023) radar system measurements of neutral winds will be a powerful tool to monitor the neutral atmospheric momentum and energy transfer in the MLT. Future plans will involve obtaining the values for the neutral atmospheric density and temperature rather than only variations from the multi-year mean. This can be accomplished through the inclusion of ablation models and further analysis (e.g., Limonta et al., 2020).

As these meteor head echo measurements are ongoing and analysis of 2024 will commence soon, it is expected that the large geomagnetic storms that have occurred in the past year will show noticeable fluctuation signatures in the MLT density. In addition, as the dataset is continually growing the background model will only improve with an increase in the number of measurements. This dataset and method of analysis is a valuable addition to the methods by which to measure the neutral atmosphere in the MLT region.

*Data availability.* The data can be accessed at the link provided to reviewers. The link and DOI will be finalized and added to the references if accepted for publication.

*Author contributions.* DH, JV, and BG conceptualized the study. RL, TR, MZ, and JLC operated the MAARSY radar and managed the data storage and distribution. JV processed the MAARSY meteor head echo data. DH, JV, and BG discussed and processed the meteor head echo mean model and variation subtraction. CCS provided the UA-ICON model run data. CCS, JFC, and JLC provided discussions on the neutral dynamics that can be detected with the technique. DH, JV, TR, DK, JK, and JLC discussed the meteor head echo processing and results. All authors contributed to the writing, editing, and reviewing of the manuscript.

*Competing interests.* One of the authors is a member of the editorial board of journal AMT.



*Acknowledgements.* DH was funded during a portion of this study through a UiT The Arctic University of Norway contribution to the

EISCAT_3D project funded by Research Council of Norway through research infrastructure grant 245683. We thank Carsten Schult, who

developed the real-time detection procedure of meteor-head echoes used in this work.





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
