# Peer review of "Monitoring of Lower Thermospheric Neutral Density Variations Using Meteor Head Echoes"

_EGUsphere, 2025_

## Author Comment (AC2)

**Dear Dr. Joel Younger,**

**We thank you for your helpful comments and suggestions. Please find responses to each item below in bold.**

**Best regards,**
**The Authors**
* * *
Reviewer 1

This paper describe the use of a large data set of radar echoes from meteor head ionization to infer density fluctuations at altitudes around 90-110 km. The work is well explained and presented and adequately cited. Of particular interest is the authors' use of velocity cubed as a proportional proxy for atmospheric density. The writing is of an overall high quality and figures are both easy to read and complimentary to the text. The paper provides a useful introduction to the authors' methodology and will serve as a useful foundation for their future publications on the topic. It is recommended for publication with the following minor changes.

General: This is perhaps pedantic, but "bulk density" may be more appropriate than "neutral density" While the atmosphere is relatively lightly ionized in the meteor ablation region, collisional heating at entry speeds does not distinguish between neutral molecules and those missing a full complement of electrons. It is recognized that "neutral density" is commonly used in literature, but this may be an opportunity to use more precise language.

**AC: We agree that both ionized and neutral atmospheric constituents will play a role. That stated, the neutral density is the heavily dominant factor. To be consistent with previous literature, we have decided to keep 'neutral density'.**

Line 20-21: It would also be worth mentioning Yi et al. 2018 (doi: 10.1002/2017JA025059)

**AC: The reference has been added**

Line 27 change "signal" to "line of sight vector" or similar

**AC: Changed**

Should define MST acronym at first use in main text.

**AC: The acronym has now been defined.**

Line 54: remove "A", use commas

**AC: Changed**

Line 84-85: It would be good to also cite the work of Campbell-Brown in generating precise maps and models of the sporadic background e.g. Campbell-Brown et al. 2008 (doi: 10.1016/j.icarus.2008.02.022)

**AC: Added the citation**

Line 85: Maybe use autumn instead of fall to avoid confusion.

**AC: Changed to autumn**

Line 103/figure 2: Is this mean detection altitude or mean initial detection altitude?

**AC: This is mean initial detection altitude. It has been clarified in the text.**

Line 110a/section 4: This assumes that the size distribution and composition of meteoroids remains constant throughout the year. While this may be a reasonable assumption for the sporadic sources, it is less so for shower sources. This raises the possibility of transient contamination of the results during strong shower activity.

**AC: This is true. We have clarified this in the text. This is also related to the comment on Figure 2 below – during meteor showers the trajectory and composition of the population will be more uniform, resulting in some contamination or bias. This is why the same DOY is analyzed as a tracer for the neutral density variations.**

Line 110b: Equation 1 describes the energy balance of collisional heating, radiation, and vaporization, but does not describe the amount of plasma generated or the reflectivity near the meteoroid. Readers would benefit from an expression describing meteor head plasma density and echo strength at the wavelengths considered. It also seems important to mention the aspect sensitivity of meteor head echoes, which may affect the average seasonal results. This only affects the absolute terms in figures 1 and 2, but not the later figures, which portray relative fluctuations between years.

**AC: We have added some discussion on the meteor head echo plasma and radar echo strength. The aspect angle is not expected to have a significant effect on the meteor head echo detections (e.g., Kero et al. 2008). It is also generally accepted that that majority of the plasma generated in the vicinity of the meteor is from the ablated meteoroid material (currently mentioned on lines 75-77 and references Dimant and Oppenheim, 2017; Sugar, 2018).**

**Kero, J., C. Szasz, G. Wannberg, A. Pellinen-Wannberg, and A. Westman (2008), On the meteoric head echo radar cross section angular dependence, *Geophys. Res. Lett.*, 35, L07101, doi:10.1029/2008GL033402.**

**Dimant, Y. S. and M. M. Oppenheim (2017), Formation of plasma around a small meteoroid: 1. Kinetic theory, *J. Geophys. Res. Space Physics*, 122, 4669–4696, doi:10.1002/2017JA023960.**

**Sugar, G., Oppenheim, M. M., Dimant, Y. S., & Close, S. (2018). Formation of plasma around a small meteoroid: Simulation and theory. *Journal of Geophysical Research: Space Physics*, 123, 4080–4093. https://doi.org/10.1002/2018JA025265**

Line 114: Would substitute "plasma density" for "ablation rate". The latter refers to material loss rate, not specifically generation rate of detectable plasma

**AC: We have made the change.**

Figure 2: These are well constructed and easy to read plots. The correlation in the top panel is clear, but the deviation of the 60 and 40 km/s heights from associated iso-density contours in the bottom panel for the first half of the year goes unremarked in the text, except the disclaimer in lines 108-

109. Do the authors think that this could be a shortcoming of the MSIS model, dynamical features of the atmosphere, or something to do with temporal changes in head echoes?

**AC: It could be due to multiple different factors. One is meteor showers, where the Eta Aquariids could be affecting the results as they have relatively large velocities. There could also be effects from the atmosphere, such as sudden stratospheric warmings.**

Line 120: "…panel of the background…"

**AC: Changed**

Line 129: incomplete sentence ending in "…to compare with."

**AC: Corrected the sentence**

Section 4.4: The summer/autumn reduction in density above 100 km is not obvious to me in figure 3. Is there some other way of presenting the data to make this claimed feature stand out?

**AC: We have added Histograms of the variation values for days 25-125 and 225-325, separated into the altitudes of 90-100 km and 100-110 km (Figure below). The figure shows a clear change in the density variations above and below 100 km for the second half of the year. We consider that this is due to a reduction in variations in the atmospheric density between years at these altitudes that affect the meteor ablation rates, and could be related to the turbopause altitude. More investigation is required to say definitively why this change in density variation occurs.**

[Figure]

Section 4.5: This section could benefit from the inclusion of a wavelet spectrogram that should clearly show the presence of planetary waves. Alternatively, a line plot showing the velocity cubed ratio variation at a fixed height may provide readers with a clearer depiction of oscillations.

**AC: We have added a further figure showing the planetary wave activity with the zoomed in figure below.  A wavelet analysis is beyond the scope of the current manuscript, but it is something planned in the future.**

---

## Author Comment (AC3)

**Dear Reviewer 2,**

**Thank you for your time in reviewing the manuscript. Please find responses to your comments below in bold.**

**Best regards,**
**The Authors**
* * *
Reviewer 2

This manuscript describes a statistical method for deriving neutral density variations in the mesosphere and lower thermosphere from 1.4 million meteor head echoes observed by the MAARSY HPLA radar between 2016 and 2023. The authors report fluctuations of 20–40% with a 3-day temporal and 2 km altitude resolution, consistent with atmospheric model predictions and influenced by geomagnetic and atmospheric events.

A fundamental concern, however, is that the work falls outside the scope of Atmospheric Measurement Techniques. The journal is explicitly dedicated to advances in measurement methodologies, including the development, intercomparison, validation, or simulation of remote sensing, in situ, and laboratory techniques. This manuscript does not present any such advancement. Rather, it applies an established radar technique to derive neutral density variations, without contributing innovation in measurement methodology, error analysis, or instrument simulation. While the scientific topic may be of interest, the absence of methodological novelty or development renders the work misaligned with the stated aims and objectives of the journal.

**AC: This comment has been addressed in a previous reply (included at the end of this document). We consider this work to be within the scope of Atmospheric Measurement Techniques, with a focus on "techniques of data processing for information retrieval for the atmosphere".**

Equally serious is the manuscript's reliance on an overly simplistic assumption that meteor detection altitude depends solely on $V^3$. The authors themselves acknowledge this limitation in the introduction:

"With this technique the height variations are determined as an average quantity with a neutral density isocontour assumed to follow this altitude variation. This provides a general overview of atmospheric neutral density variations, but provides minimal information about differences between altitudes for the same time."

Such an assumption is physically unsound. A more rigorous treatment would account for kinetic energy, since the ablation profile—and thus detection altitude—of a large, slow particle may closely resemble that of a small, fast one. Furthermore, the astronomical origin of the meteoroids, and therefore the entry angle, exerts a significant influence on detection altitude, as do local atmospheric conditions (e.g., Dawkins et al., 2024). Equally, the physical composition of the meteoroids has been shown to play a decisive role in ablation behaviour in optical studies (e.g., Kikwaya et al., 2011a,b). That the authors chose not to employ a comprehensive ablation model, despite the ready availability of such tools, is a serious shortcoming that undermines the robustness of the presented analysis.

**AC: We do not assume that the meteor detection altitude depends solely on V^3, we assume that statistically the other factors should be similar for a population of meteors measured on the same day-of-year (DOY). Therefore the primary difference for the same DOY corresponds to atmospheric density.**

**Many of the factors listed in the comment consider a single meteor event, but here we consider the statistical population of meteors. Factors such as the entry angle and the composition of meteors should be relatively consistent statistically for the same DOY between years. It is due to the numerous meteor head echo events made available with this MAARSY dataset that it is possible to perform the analysis in this study.**

In addition to these two fundamental flaws, there are numerous further issues which must be addressed:

Throughout the manuscript the authors refer to "measuring the neutral density". This is incorrect; the study concerns variability in neutral density, which is conceptually distinct.

**AC: We agree that we should clarify that we are determining neutral density variations, and have made the corresponding changes to the text.**

Page 3, line 34: The claim that "Measurements of the meteor head plasma provide more details on the meteor ablation and trajectory, but require HPLA radars" is outdated. This has not been true for some time (see Janches et al., 2014; Panka et al., 2021).

**AC: We have rephrased this to "… but require HPLA radars to consistently make measurements of microgram sized ablating meteoroids, which greatly increases the number of meteor head echo detections."**

Page 3, line 77: The statement "This suggests that meteoroids with higher velocity are, on average, detected at higher altitudes" is not a suggestion—it has long been established (e.g., Janches & ReVelle, 2005; Vondrak et al., 2008). Moreover, as noted earlier, detection altitude depends on several other parameters, particularly in the context the authors are attempting to present.

**AC: We have removed that it is suggested.**

Page 3, line 84: The authors state that "Throughout the year, the radiant distribution of meteors observed by a radar changes (e.g., Janches et al., 2006; Kero et al., 2012)." This is correct but incomplete. The variability is highly location dependent; at equatorial sites, for example, such changes are minimal.

**AC: We have added that the variability is location dependent.**

Page 5, lines 95–100: The discussion presented is already well established in the literature, and the variability again depends strongly on geographical location. For example, such variability has not been measured at equatorial latitudes (see Sparks & Janches, 2009a,b). The authors should clarify and reference the prior work properly.

**AC: We have added a reference and included some discussion about variability being location dependent.**

Page 5, line 102: The repeated reference to the "background model" is ambiguous. It is not clear what background the authors are referring to, and this requires clarification.

**AC: We have clarified what is referred to by the "background model".**

In its present form, the manuscript suffers from serious conceptual, methodological, and contextual shortcomings. Most importantly, it does not offer an advance in measurement methodology and therefore lacks relevance to Atmospheric Measurement Techniques. The paper may be more appropriately considered by a journal focused on atmospheric dynamics or variability, rather than one dedicated to the advancement of measurement techniques.

**AC: We respectfully disagree. As we have stated, this study uses techniques of data processing for information retrieval for the atmosphere, which falls within the scope of Atmospheric Measurement Techniques. The applied technique to this dataset is novel, and provides a new method of tracking neutral density variations as a function of altitude between years. This technique will provide new opportunities for scientific studies, some of which are briefly discussed in the manuscript.**

References

Dawkins E. C., D. Janches, G. Stober, et al. 2024. "Seasonal and Local Time Variation in the Observed Peak of the Meteor Altitude Distributions by Meteor Radars." Journal of Geophysical Research: Atmospheres 129 (21): [10.1029/2024jd040978] [Journal Article/Letter]

Janches D. and D. O. ReVelle. 2005. "The Initial Altitude of the Micrometeor Phenomenon: Comparison between Arecibo radar observations and theory." Journal of Geophysical Research 110 A08307 [10.1029/2005JA011022] [Journal Article/Letter]

Janches D., W. Hocking, S. Pifko, et al. 2014. "Interferometric meteor head echo observations using the Southern Argentina Agile Meteor Radar." Journal of Geophysical Research - Space Physics 119 [10.1002/2013JA019241] [Journal Article/Letter]

Kikwaya, J.-B., Campbell-Brown, M., & Brown, P. 2011,406 Astronomy & Astrophysics, 530, A113,407 doi: 10.1051/0004-6361/201116431408

Kikwaya, J. B., Campbell-Brown, M., & Brown, P. G. 2011, A&A, 530, A113, doi: 10.1051/0004-6361/201116431

Panka P. A., R. J. Weryk, J. S. Bruzzone, et al. 2021. "An Improved Method to Measure Head Echoes Using a Meteor Radar." The Planetary Science Journal 2 (5): 197 [10.3847/psj/ac22b2] [Journal Article/Letter]

Sparks J. J. and D. Janches. 2009a. "Latitudinal dependence of the variability of the micrometeor altitude distribution." Geophysical Research Letters 36 L12105 [10.1029/2009GL038485] [Journal Article/Letter]

Sparks J. J. and D. Janches. 2009b. "Correction to ''Latitudinal dependence of the variability of the micrometeor altitude distribution''." Geophysical Research Letters 36 L17101 [10.1029/2009GL039987] [Journal Article/Letter]
* * *
**Initial Response:**

**Dear Reviewer,**

**Thank you for your time in reviewing the manuscript. We wish to respond to your comment that this work is not within the scope of AMT. We respectfully disagree.**

**While the measurement of meteor head echoes is established, here we leverage the extensive dataset available to showcase a novel analysis technique applied to meteor head echoes to derive atmospheric neutral density variations between years for the same day-of-year. To quote the AMT landing page (https://www.atmospheric-measurement-techniques.net/), "The main subject areas comprise the development, intercomparison, and validation of measurement instruments and techniques of data processing and information retrieval for gases, aerosols, and clouds." The work we present falls well within the subject of techniques of data processing for information retrieval for the atmosphere.**

**We also wish to emphasize that the methodology presented has not been applied previously to meteor head echo measurements. We are able to perform this analysis due to the extensive dataset of greater than 1 million meteor head echo detections made with the MAARSY radar system on a consistent basis between the years of 2016-2023. To the authors knowledge, no other meteor head echo dataset of this magnitude is available globally.**

**The remaining comments will be addressed during the response phase.**

**Best regards,**
**Devin Huyghebaert**